# OpenReview forum: "REINFORCE Adversarial Attacks on Large Language Models: An Adaptive, Distributional, and Semantic Objective"
_ICML.cc/2025/Conference — ICML 2025 poster_

### Official Review · Reviewer_hpiv · 2025-03-05

**Overall Recommendation:** 3

**Summary:**

The paper demonstrates that existing LLM jailbreak defenses significantly underestimate model vulnerability due to non-adaptive attack objectives. By adopting a reinforcement learning-based approach, adversarial attacks can become more effective and adaptive, posing a greater challenge for safety alignment efforts. The authors suggest that adaptive attack objectives should be the standard for future robustness evaluations of LLMs.

**Claims And Evidence:**

Claim: Existing jailbreak attacks fail to capture true model vulnerability as they do not adapt to model-specific responses.
Evidence: Figure 2 shows that affirmative-response attacks often produce harmless completions, whereas REINFORCE successfully generates harmful responses.

Claim: The proposed method achieves substantially higher ASR than standard attacks.
Evidence: Llama 3 8B: ASR 26% → 68% (Table 1). Circuit-breaker defense: ASR 2% → 22% (Table 3). Further improvement to 61% ASR with better attack seeding.

Claim: The attack optimizes the full response distribution, not just a fixed affirmative phrase.
Evidence: Section 2 formulates the attack as reinforcement learning, optimizing expected harmfulness rather than likelihood maximization.


Claim: REINFORCE can enhance multiple jailbreak attack methods.
Evidence: REINFORCE-GCG and REINFORCE-PGD consistently outperform their baselines (Tables 1 & 2). The sampling strategy adapts dynamically, leading to more effective attacks.

**Essential References Not Discussed:**

N/A

**Experimental Designs Or Analyses:**

The paper presents a well-structured experimental design to evaluate the effectiveness of the proposed REINFORCE-based adversarial attack. The paper benchmarks against state-of-the-art attack methods (GCG and PGD) using the standard affirmative-response objective. It evaluates both non-defended and defended LLMs (e.g., Llama 3 with circuit breakers). While comparisons with existing attacks are robust, no experiments compare against non-gradient-based attacks, such as those using generative models (e.g., adversarial LLM-generated prompts). The reliance on greedy decoding alone ignores the stochastic nature of LLM responses.

**Methods And Evaluation Criteria:**

The methods and evaluation criteria are well-chosen for the problem, ensuring strong empirical comparisons. However, broader comparisons (e.g., against LLM-generated attacks) and alternative evaluation strategies (e.g., human judges or diverse decoding methods) could strengthen the robustness of the findings.

**Other Comments Or Suggestions:**

N/A

**Other Strengths And Weaknesses:**

See above

**Questions For Authors:**

See above

**Relation To Broader Scientific Literature:**

See above

**Theoretical Claims:**

The paper presents several theoretical claims related to adversarial attacks on LLMs, framed through reinforcement learning (RL) and distributional optimization.
Eq. (1) defines the adversarial objective as maximizing the expected reward over responses.
Eq. (6) establishes equivalence between this optimization and the RL value function.
Eq. (7) applies the policy gradient theorem, showing that optimizing for harmful outputs can be done using REINFORCE.
The theoretical claims are correctly derived and well-supported by standard reinforcement learning principles. While the mathematical formulations are sound, the paper lacks formal guarantees on sample efficiency, judge reliability, and convergence. These aspects could be explored in future work.

---

> ### Author Rebuttal · Authors · 2025-04-01
>
> We thank the reviewer for the suggestion and plan to investigate theoretical guarantees in future work.
>
> ## Alternative evaluation strategies
>
> While our objective might utilize false positives of the judge instead of triggering actually harmful behavior, such cases clearly do not appear systematically in our experiments. The reader can check this by going through the examples provided (Appendix D and E). We will extend our statement in the right column of lines 296-298 to better convey this point and revise the limitations (Section 6, lines 386-394, right column).
>
> ## Further baselines
>
> Due to the policy-gradient approach, the evaluation of our objective focuses on gradient-based attacks. Additionally, HarmBench's results show that GCG is superior in terms of ASR to non-gradient-based attacks that, e.g., use a generative model. To the best of our knowledge, no other attack has been shown to be considerably stronger than GCG on HarmBench, which we consider to be a state-of-the-art jailbreak benchmark. Hence, we did not include further comparisons in the submission since these baselines would certainly perform worse than GCG with affirmative objective. Nevertheless, we consider following the suggestion in a revised version of the paper.
>
> ## Greedy decoding
>
> We agree that the reliance on greedy decoding has limitations (e.g., see Scholten et al. 2024, as referenced in our paper). However, the greedy evaluation is the default (GCG, HarmBench, ...), and except for a handful of exceptions, virtually all papers studying LLM jailbreaks rely on the greedy evaluation. Hence, we have decided to stick to the convention of evaluating the greedy generation for comparability to other works. We will consider adding an experiment in a revised version of the paper studying this distributional perspective.
>
> We kindly ask for clarification on any remaining concerns.

---

### Official Review · Reviewer_WqKi · 2025-03-08

**Overall Recommendation:** 3

**Summary:**

This paper addresses the challenge of jailbreaking large language models (LLMs) – i.e. crafting adversarial prompts that make an aligned (safety-trained) model produce disallowed or harmful content. The authors point out a key limitation in current adversarial prompt attacks: they typically optimize a static objective such as maximizing the likelihood of a particular “affirmative response” (a fixed harmful reply prefix). This static objective doesn’t adapt to the attacked model’s actual behavior and treats the model output as if it were a single target sequence​.  As a result, prior attacks often succeed in forcing the model to begin with a harmful-looking phrase, yet the model may still refuse or derail thereafter (yielding a benign completion). Such non-adaptive attacks can overestimate the model’s robustness because a high likelihood of a fixed trigger phrase doesn’t guarantee a genuinely harmful outcome.

The paper proposes REINFORCE Adversarial Attacks, a novel adaptive, distributional, and semantic optimization objective for generating adversarial prompts. Instead of focusing on one predetermined “bad” answer, their method explicitly optimizes the expected harmfulness of the model’s entire output distribution. They cast adversarial prompt search as a reinforcement learning (RL) problem: the prompt is treated as a policy (initial state) that influences the distribution of outputs, and the goal is to maximize a reward measuring harmful content in the output.

The authors integrate their RL-based objective into two state-of-the-art jailbreak attack algorithms: Greedy Coordinate Gradient (GCG)​ and Projected Gradient Descent (PGD) attacks. Empirical results show that the proposed REINFORCE-based objective leads to substantially higher attack success rates (ASR) compared to the conventional static objective.

**Claims And Evidence:**

Claim 1: Static “affirmative response” objectives are flawed and non-adaptive, leading to overly optimistic estimates of robustness. The authors assert that existing attacks which maximize the likelihood of a fixed target response do not adequately test a model’s true vulnerabilities. They support this claim with a compelling anecdotal example (Fig. 2 and accompanying text) where the baseline attack indeed finds a prompt that makes the model start with the desired forbidden phrase (“Sure, here’s how to…”) yet the model’s continuation is not actually harmful​.

Claim 2: The proposed REINFORCE-based objective is adaptive, distributional, and optimizes the true probability of harmful outputs. The authors devote Section 2–3 to formalizing this claim. They treat finding an adversarial prompt x̃ as maximizing the expected reward $E_{y\sim P_\theta(\cdot|x̃)}[ \text{Reward}(y,x̃) ]$​.

Claim 3: The new attack objective yields substantially higher attack success rates (ASR) on current LLMs, revealing greater vulnerability. This is an empirical claim, and the paper provides strong experimental evidence to support it. In Tables 1–2, results on five different models consistently show large ASR gains with the REINFORCE objective compared to the baseline “Affirmative” objective.

On the whole, the paper’s claims are well-supported. The combination of theoretical justification, quantitative results, and qualitative examples makes for a convincing argument. The only slight gap is that the paper doesn’t deeply analyze why one model (Llama 2 7B under PGD) didn’t improve.

**Essential References Not Discussed:**

See above.

**Experimental Designs Or Analyses:**

The experimental design is thorough and the analyses are generally sound, successfully supporting the paper’s conclusions.

1. Model Selection and Generality: The authors tested their attacks on a diverse set of models (five different LLMs plus a defended variant). This breadth is commendable as it demonstrates the attack’s robustness across model families and sizes.
2. Consistency and Repetition: For each model, they evaluate on 50 prompts with both the baseline and new attack, ensuring a direct side-by-side comparison. Because each prompt attack is quite involved, they didn’t do repeated trials on the same prompt (which could measure stochastic variance). However, given the large improvements, it’s unlikely that variance would overturn their conclusions.
3. Outlier Analysis: The only case where the new attack did not outperform was PGD on Llama2 7B (18% vs 18% ASR)​. The authors highlight this as “the only exception”​. They don’t provide a deep analysis in the paper about why it remained unchanged. It raises curiosity: Llama2 7B did see improvement under GCG (38→62%)​, so why would PGD not improve?
4. Ablation and Sensitivity: The paper does not present extensive ablation studies on the components of their approach (like the effect of sample size K, or the biased sampling strategy vs purely random sampling, etc.). They mention in Appendix C some adjustments (e.g. not using the random sample for candidate selection in GCG to save time)​, which implies they tried variants for efficiency. This suggests they did some hyperparameter search for things like the number of samples K, the inclusion of $y_{\text{seed}}$, etc., on separate prompts. It would be interesting to see those ablation results, but they are not included (likely due to space or because they felt it was straightforward). The absence of detailed ablation does not critically harm the paper—the main narrative is well-supported by the straightforward baseline vs. new comparisons. However, it leaves some questions unanswered, like how important is the biased sampler or how sensitive is the attack to the initial seed prompt. The authors do give one piece of analysis in the circuit-breaker experiment: by changing $y_{\text{seed}}$ from the affirmative phrase to a more harmful one (from a successful base model attack), they dramatically improved results (22%→61%)​.

**Methods And Evaluation Criteria:**

The experimental methodology is solid. The use of a strong benchmark (HarmBench), multiple models, and direct comparison to known attack baselines makes the results meaningful. The authors were careful to keep comparisons fair (same hyperparameters, etc.) and to document any deviations. The evaluation criteria (ASR via a judge model) is appropriate for the task and was applied uniformly. One might suggest minor improvements, like including more prompts or multiple random restarts to measure variance, but given the resource-intensive nature of these attacks, the choices made are quite reasonable. The evidence provided – in the form of tables and examples – is directly tied to the stated methods and metrics. Overall, the methods and evaluation are well-aligned with the problem of jailbreaking LLMs, and they credibly demonstrate the value of the proposed approach.

**Other Comments Or Suggestions:**

N/A

**Other Strengths And Weaknesses:**

1. Clarity of Problem Statement: The paper clearly identifies a concrete problem – the static nature of existing jailbreak objectives – and explains it in intuitive terms (the example of the model just because it said “Sure, …” doesn’t mean it actually gave a harmful answer). The introduction uses simple language to explain why non-adaptive attacks can be misleading.
2. Computational Intensity and Practicality: One weakness is that the method, as presented, is computationally expensive and requires white-box access.
3. Lack of Defense or Mitigation Discussion: The paper is focused on attacks and doesn’t propose any defenses or mitigations.
4. Missing Discussion on Judge Robustness: As mentioned, the method’s success hinges on the judge. If the model found a “trick” to say something harmful in a way the judge doesn’t recognize, the attack might succeed from a human perspective but not be counted. Or vice versa, it might fool the judge into thinking a safe response is harmful (less likely, but possible). The authors did not report any instances of misclassification by the judge (which suggests the judge did well). A weakness is that they did not explicitly validate the judge’s decisions with human oversight (perhaps assuming HarmBench did that).

**Questions For Authors:**

N/A

**Relation To Broader Scientific Literature:**

This paper sits at the intersection of adversarial machine learning, natural language generation, and AI safety. Its contributions should be viewed in the context of several lines of prior work: LLM jailbreaks/adversarial prompts, adversarial training/evaluation in NLP, and controlled text generation via optimization. The authors do a good job situating their work among recent studies, though there are a few older relevant works that could also be acknowledged.

1. Advances over Prior Jailbreak Attacks: The authors reference numerous recent papers on jailbreaking or prompt attacks for LLMs​, including Zou et al. (2023), Perez et al. (2022), Wen et al. (2023), Liu et al. (2024), Zhu et al. (2023), Geisler et al. (2024), Guo et al. (2024). These works collectively indicate a surge of interest in automatically finding prompts that cause misbehavior. For instance, Zou et al. (2023) introduced the GCG method used as a baseline here, and Geisler et al. (2024) introduced the PGD attack – both are cited and directly built upon.
2. Connection to Adversarial ML (Robustness Evaluation): The concept of adaptive vs non-adaptive attacks is well-known in the adversarial ML literature for classification models. The authors reference Carlini & Wagner (2017)​, who famously pointed out that defenses must be evaluated against adaptive attacks (where the attacker knows the defense), otherwise one can get a false sense of security. However, the connection to older adversarial attacks (like those in computer vision) is not discussed.
3. Comparison to Prior Results: The paper doesn’t explicitly compare its results to prior jailbreak success rates from other papers (except the ones they re-implemented like GCG, PGD).
4. Connection to RLHF: the REINFORCE algorithm is also used in RLHF for LLMs. However, these works are not mentioned in the paper.

**Theoretical Claims:**

This paper is mainly empirical and does not provide theoretical results for the main algorithms (PGD and GCD) proposed.

---

> ### Author Rebuttal · Authors · 2025-04-01
>
> ## Mitigation
>
> The robustness literature suggests that only systematic methods like adversarial training actually help. For adversarial training, the attack effectivity is key for actual improvements (e.g., see [Kolter and Madry, 2018](https://adversarial-ml-tutorial.org/adversarial_training/) arguments via Danskin's theorem). Hence, powerful attacks should translate to powerful adversarial training and an effective mitigation strategy. We will add a discussion.
>
> ## More prompts
>
> Following the suggestion, we following report results on 200 prompts (instead of 50 prompts)
>
> For GCG:
> |  | Affirmative | REINFORCE *(ours)* |
> |---|:---:|:---:|
> | Gemma 1.1 2B | 0.57 | **0.88** |
> | Gemma 1.1 7B | 0.63 | **0.87** |
> | Llama 2 7B | 0.32 | **0.56** |
> | Llama 3 8B | 0.35 | **0.73** |
> | Vicuna 1.5 7B | 0.86 | **0.95** |
>
> For PGD:
> |  | Affirmative | REINFORCE *(ours)* |
> |---|:---:|:---:|
> | Gemma 1.1 2B | 0.56 | **0.82** |
> | Gemma 1.1 7B | 0.54 | **0.84** |
> | Llama 2 7B | 0.17 | **0.22** |
> | Llama 3 8B | 0.57 | **0.69** |
> | Vicuna 1.5 7B | 0.87 | **0.94** |
>
>
> ## Confidence intervals
>
> In relation to the suggestion, we will include Clopper-Pearson intervals to show statistical significance. For example, for GCG on Gemma 1.1 2B, the 90%-confidence intervals are [0.49, 0.63] (affirmative) vs. [0.82, 0.92] (REINFORCE).
>
> ## Ablation
>
> We did not include ablations that we thought were not insightful. For example, excluding the random sample for candidate selection did not impact the performance much.
>
> Regarding the sampling, we refer to the ablation study in Table 4. We did not experiment with more samples than we have in our experiments, except for including the initial greedy response, which did not help.
>
> To study the impact of $\mathbf{y}_{\text{seed}}$, we ran experiments using the concurrent/very recent AdvPrefix (Zhu et al., 2024) using GCG and reporting ASR@512:
>
> |  | Affirmative | Affirmative | REINFORCE *(ours)* | REINFORCE *(ours)* | REINFORCE *(ours)* |
> |---|:---:|:---:|:---:|---:|---:|
> | $\mathbf{y}_{\text{seed}}=$ | $\mathbf{y}_{\text{affirmative}}$ | $\mathbf{y}_{\text{advprefix}}$ | $\mathbf{y}_{\text{affirmative}}$ | $\mathbf{y}_{\text{advprefix}}$ | $\mathbf{y}_{\text{history}}$ |
> | Llama 3 8B | 0.35 | 0.70 | 0.73 | **0.81** | - |
> | + Circuit breaker | 0.02 | 0.14 | 0.23 | 0.48 | **0.50** |
>
> $\mathbf{y}\_{\text{affirmative}}$ is HarmBench's target, and $\mathbf{y}\_{\text{history}}$ the generation of a previously successful attack on Llama 3 8B w/o defense. Having a better seed $\mathbf{y}\_{\text{seed}}$ clearly helps. However, our REINFORCE objective further reinforces attack efficacy.
>
> ## Further citations
>
> We are happy to incorporate further references.
>
> Due to the vast body of robustness literature, we would appreciate further pointers for particularly relevant "older" works.
>
> We do reference Ahmadian et al., 2024 (REINFORCE for RLHF) but are happy about further pointers to relevant works.
>
> ## Comparison to Prior Results
>
> Due to the policy-gradient approach, the evaluation focuses on gradient-based attacks. Additionally, HarmBench's results show that GCG is superior in terms of ASR to non-gradient-based attacks that, e.g., use a generative model like PAIR. To the best of our knowledge, no other attack has been shown to be considerably stronger than GCG on HarmBench (state-of-the-art jailbreak benchmark). Hence, these baselines would even perform worse than GCG with affirmative objective.
>
> ## Computational cost
>
> While each attack step is more expensive, our REINFORCE-GCG obtains a better ASR-runtime tradeoff (e.g., Figure 3). Hence, our REINFORCE-GCG either achieves the same ASR in less time or obtains a higher ASR, given equal compute. We think it is a promising direction for future work to further study techniques for lowering the computational cost. Some ad hoc strategies could be speculative decoding (Leviathan et al., 2023) or tree-based attention (Cai et al., 2024) to avoid duplicate computations. For a better overview, we will add detailed breakdowns of the time cost of REINFORCE-GCG.
>
> ## White-box access
>
> In Figure 4, we investigate an application of our objective *without using gradient information*. Similarly to other works (e.g., Andriushchenko et al.), we instead apply uniformly random perturbations and then select the best candidate. From the dashed blue bar with the solid blue bar, it follows that also attacks w/o gradient information benefit from our objective.
>
> ## Judge Robustness
>
> We agree that "reward hacking" is one of the potential drawbacks. Thus, we include random responses that are deemed harmful in Appendix D and E. While our objective might utilize false positives of the judge instead of triggering actually harmful behavior, such cases clearly do not appear systematically in our experiments. The reader can check this by going through the examples provided. We will extend our paper in that regard.
>
> We kindly ask for clarification on any remaining concerns.

---

### Official Review · Reviewer_G8az · 2025-03-10

**Overall Recommendation:** 4

**Summary:**

The paper "REINFORCE Adversarial Attacks on Large Language Models: An Adaptive, Distributional, and Semantic Objective" presents a novel approach for adversarial attacks on large language models (LLMs). Traditional optimization-based adversarial attacks rely on maximizing the likelihood of a predefined affirmative response, which often does not translate to truly harmful completions. The authors introduce an adaptive and semantic optimization approach that leverages the REINFORCE policy-gradient method. This approach optimizes over the distribution of model responses rather than relying on a static target. The proposed method significantly improves attack success rates for jailbreak algorithms like Greedy Coordinate Gradient (GCG) and Projected Gradient Descent (PGD), demonstrating its efficacy in evading safety mechanisms in LLMs. The paper provides extensive empirical validation, showing that the REINFORCE objective enhances attack success rates, including against the circuit breaker defense in Llama 3, increasing the ASR from 2% to 50%.

**Claims And Evidence:**

The primary claims made by the paper are:

The affirmative response objective is inconsistent and can lead to overestimated robustness.

The proposed REINFORCE objective is adaptive, distributional, and semantic, making it more effective for adversarial attacks.

The method significantly improves attack success rates for existing jailbreak algorithms (GCG and PGD).

The REINFORCE objective successfully bypasses state-of-the-art safety mechanisms, including circuit breakers in Llama 3.

These claims are well-supported by empirical results. The authors provide detailed comparisons showing that their method consistently outperforms baseline attacks across various LLMs. The increase in ASR for models such as Llama 3 8B (from 35% to 73%) and with the circuit breaker defense (from 2% to 50%) strongly supports their assertions.

**Essential References Not Discussed:**

The paper cites most relevant works in adversarial robustness and jailbreak attacks. However, additional discussion on interpretability and mitigation strategies for adversarial prompts could further contextualize the contributions.

**Experimental Designs Or Analyses:**

The experimental design is robust, with evaluations conducted on multiple LLMs, including Llama 2, Llama 3, Gemma, and Vicuna. The use of diverse benchmarks and comparative analysis with existing jailbreak techniques strengthens the findings. However, some areas, such as sensitivity to hyperparameters and different attack settings, could be explored further.

**Methods And Evaluation Criteria:**

The proposed method uses reinforcement learning (specifically the REINFORCE algorithm) to optimize adversarial prompt crafting. The evaluation criteria include:

Attack Success Rate (ASR), measured across multiple LLMs.

The effectiveness of attacks against standard and advanced defenses.

Comparison with state-of-the-art jailbreak methods (GCG and PGD).

Ablation studies to analyze the impact of different sampling strategies.

The chosen evaluation benchmarks (e.g., HarmBench) and experimental setups are appropriate for assessing the effectiveness of adversarial attacks.

**Other Comments Or Suggestions:**

It would be beneficial to explore the implications of these attacks on commercial models with additional safety guardrails.

Further discussion on ethical considerations and responsible disclosure of adversarial methods would strengthen the paper.

Future work could examine real-world deployment scenarios and adaptive defenses against REINFORCE-based attacks.

**Other Strengths And Weaknesses:**

Strengths:

Introduces an innovative, theoretically grounded attack objective.

Demonstrates significant improvements over existing jailbreak methods.

Provides extensive empirical validation across multiple models and defenses.

Strong methodological rigor with reinforcement learning integration.

Weaknesses:

The reliance on LLM-as-a-judge evaluations may introduce biases in measuring attack success.

Limited discussion on potential mitigations for adversarial attacks.

Computational overhead for the REINFORCE optimization process.

**Questions For Authors:**

How does the computational cost of REINFORCE-based attacks compare to standard jailbreak methods in real-world scenarios?

Have you tested the method on closed-source models like OpenAI GPT-4 or Claude to assess generalizability?

How sensitive is the attack success rate to hyperparameter tuning in REINFORCE optimization?

What countermeasures do you propose for mitigating the effectiveness of your attack method?

Could the REINFORCE framework be adapted to enhance LLM safety rather than bypassing it?

**Relation To Broader Scientific Literature:**

The work aligns with existing research in adversarial attacks on LLMs and extends previous methods by incorporating reinforcement learning-based optimization. It builds upon prior works in adversarial robustness, jailbreak attacks, and policy-gradient methods. The findings are relevant to both security researchers and those working on LLM alignment and safety.

**Theoretical Claims:**

The paper presents a theoretical formulation of adversarial attacks on generative models and derives an attack objective using REINFORCE. The correctness of the mathematical formulations and their application to reinforcement learning are well-grounded in existing literature. The authors reference foundational works (e.g., Williams, 1992) to support their approach.

---

> ### Author Rebuttal · Authors · 2025-04-01
>
> We thank the reviewer for the feedback and the numerous suggestions! We will address the points made in a revised version of the paper. Next, we elaborate on some of the points and answer the questions.
>
> ## The reliance on LLM-as-a-judge evaluations may introduce biases in measuring attack success.
>
> While our objective might utilize false positives of the judge instead of triggering actually harmful behavior, such cases clearly do not appear systematically in our experiments. The reader can check this by going through the examples provided (Appendix D and E). We will extend our statement in the right column of lines 296-298 to better convey this point.
>
> ## How does the computational cost of REINFORCE-based attacks compare to standard jailbreak methods in real-world scenarios?
>
> While each attack step is more expensive, our REINFORCE-GCG obtains a better ASR-runtime tradeoff (e.g., Figure 3). Hence, our REINFORCE-GCG either achieves the same ASR in less time or obtains a higher ASR when given equal computing. We think it is a promising direction for future work to further study techniques for lowering the computational cost. Some ad hoc strategies could be speculative decoding (Leviathan et al., 2023) or tree-based attention (Cai et al., 2024) to avoid duplicate computations. For a better overview, we will add detailed breakdowns on the time cost of REINFORCE-GCG to a revised version of the paper.
>
> ## Have you tested the method on closed-source models like OpenAI GPT-4 or Claude to assess generalizability?
>
> We did not investigate attacking closed-source models due to the lack of resources. We leave such studies open for future work.
>
> ## How sensitive is the attack success rate to hyperparameter tuning in REINFORCE optimization?
>
> We study the most critical hyperparameters in Table 4, namely the used samples. Beyond that, we did not particularly tune other hyperparameters since the ASR was not very sensitive to changes in them.
>
> ## What countermeasures do you propose for mitigating the effectiveness of your attack method? \& Could the REINFORCE framework be adapted to enhance LLM safety rather than bypassing it?
>
> From the vast literature on adversarial robustness, there is limited hope that vulnerabilities can be effectively mitigated beyond systematic methods like adversarial training. For adversarial training, the attack effectivity is key for an actual improvement of the robustness (e.g., see [Kolter and Madry, 2018](https://adversarial-ml-tutorial.org/adversarial_training/) arguments via Danskin's theorem). Hence, powerful attacks should translate to powerful adversarial training and an effective mitigation strategy. We will add such a discussion in a revised version of the paper.

---

### Official Review · Reviewer_Qgsz · 2025-03-14

**Overall Recommendation:** 4

**Summary:**

The authors propose a new text-based adversarial loss function for jailbreak attacks, addressing the limitation that optimizing solely for affirmative responses (e.g., "Sure, here is how to...") can lead to non-harmful completions. To improve effectiveness, the authors introduce a loss function that incorporates multiple response samplings in the gradient computation: random sampling (low-temperature generation), greedy sampling (standard), seed-based sampling (biasing toward affirmative responses or prior successful attacks), and harmful sampling (most likely harmful response). The paper demonstrates that applying this change to existing jailbreak methods, such as GCG and PGD, improves the attack success rate.

**Claims And Evidence:**

The claims are supported by convincing evidence.

**Essential References Not Discussed:**

There are existing methods that optimize to avoid rejection responses (e.g., "I'm sorry, I cannot answer") [3], rather than directly optimizing for affirmative responses. In relation to Figure 1, these approaches cover a portion of the shaded grey area and should be discussed to provide a more comprehensive comparison of optimization objectives in adversarial attacks.

**Ethical Review Flag:**

Flag this paper for an ethics review.

**Experimental Designs Or Analyses:**

Experiment designs seem valid and sound.

**Methods And Evaluation Criteria:**

The proposed method directly addresses the identified limitation of the affirmative attack objective having blind spots. Both the approach and the analysis are well-aligned with the problem, making the methodology and evaluation criteria appropriate for the task.

**Other Comments Or Suggestions:**

No.

**Other Strengths And Weaknesses:**

Strengths:
- Clear writing that effectively communicates the methodology and findings.
- Strong mathematical foundation, providing a well-supported theoretical basis for the approach.
- Convincing experimental results demonstrating the effectiveness of the proposed method.

Weaknesses:
- Incomplete experimental settings: One key application of GCG is generating a universal adversarial suffix—does the proposed method reduce computational cost or improve attack success rate (ASR) in this setting? Additionally, while less critical, evaluating the transferability of suffixes across models would add further insight.
- Lack of ablation studies: How was the clamping value of the seed determined? An ablation study examining its impact would strengthen the empirical analysis.
- Limited comparison with recent attacks: While the experiments demonstrate improvements over GCG and PGD, a broader contextualization against more recent jailbreak attacks (e.g., [1,2,3], doesn't have to be these specifically) would provide a clearer picture of where this method stands in jailbreak performance.


References:

[1] Liu, Xiaogeng, et al. "Autodan: Generating stealthy jailbreak prompts on aligned large language models." arXiv preprint arXiv:2310.04451 (2023).

[2] Paulus, Anselm, et al. "Advprompter: Fast adaptive adversarial prompting for llms." arXiv preprint arXiv:2404.16873 (2024).

[3] Chao, Patrick, et al. "Jailbreaking black box large language models in twenty queries." arXiv preprint arXiv:2310.08419 (2023).

**Questions For Authors:**

Q1: Given that computational cost is a major limitation of gradient-based methods, how could this approach be incorporated into a non-gradient-based method like [3]? A discussion on potential adaptations or extensions would be valuable in understanding the broader applicability of this method.

**Relation To Broader Scientific Literature:**

The paper proposes a modification to the attack objective for existing gradient-based jailbreak attacks such as GCG and PGD. While recent work in this area has largely focused on enhancing attacks by adding features (e.g., improving stealth [1]) or reducing computational costs by eliminating gradient requirements [2], this method takes a different approach. It proposes a direct improvement for attack success rate by refining the optimization objective itself.

**Theoretical Claims:**

I have reviewed the mathematical formulation at a high level, and it appears correct. However, reinforcement learning formulations are not my area of expertise.

---

> ### Author Rebuttal · Authors · 2025-04-01
>
> We thank the reviewer for the thorough feedback!
>
> ## Existing methods that optimize to avoid rejections
>
> We thank the reviewer for pointing out works that avoid rejections. While we have already referenced the mentioned work, we have not explicitly discussed this alternative objective. In a revised version of the manuscript, we will include this in our discussions.
>
> ## Does the proposed method reduce computational cost or improve attack success rate (ASR) in this setting?
>
> Our REINFORCE-GCG obtains a better ASR-runtime tradeoff (e.g., Figure 3). Hence, our REINFORCE-GCG either achieves the same ASR in less time or obtains a higher ASR, given equal compute resources.
>
> ## Lack of ablation studies: How was the clamping value of the seed determined?
>
> We observed that the judge often evaluates an affirmative response as non-harmful. However, to provide guidance to the attack and to have the regular affirmative objective as a special case of our objective, we decided to clamp it to some small constant.
>
> As long as the LLM generates benign responses, the seed will dominate the guidance towards harmful behavior. Once the LLM generates harmful responses, the actual harmful responses dominate. This is especially true due to the rather binary behavior of the HarmBench judge (usually either returns ~0 or ~1). Consequently, the exact value is of minor importance, and we did not include an ablation since we thought it was not interesting enough.
>
> ## Limited comparison with recent attacks
>
> Due to the policy-gradient approach, the evaluation of our objective focuses on gradient-based attacks. Additionally, HarmBench's results show that GCG is superior to attacks [1,3] in terms of ASR. To the best of our knowledge, no other attack has been shown to be considerably stronger than GCG on HarmBench, which we consider to be a state-of-the-art jailbreak benchmark. Hence, we did not include further comparisons in the submission since these baselines would certainly perform worse than GCG with affirmative objective. Nevertheless, we consider following the suggestion in a revised version of the paper.
>
> ## Computational cost of gradient
>
> It is somewhat of a misconception that the gradient calculation was costly (in a GCG-style attack). Recall that GCG does one forward+backward pass for generating 512 mutations based on the gradient information. Thereafter, the cross entropy loss w.r.t. the affirmative objective is calculated for all 512 mutations/candidates to determine the best mutation. Thus, the gradient calculation is usually well below 5% of the total runtime.
>
> ## Usage of our objective is non-gradient-based methods
>
> In Figure 4, we investigate an application of our objective *without using gradient information*. Similarly to other works (e.g., Andriushchenko et al.), we instead apply uniformly random perturbations and then select the best candidate. From the comparison of the dashed blue bar with the solid blue bar, it is clear that other optimization-based approaches that do not use gradient information would benefit from our objective. We will discuss this more prominently in a revised version of our paper.

---

### Decision · Program_Chairs · 2025-05-01

**Decision:**

Accept (poster)

**Comment:**

This paper presents a novel adversarial attack strategy on LLMs, improving upon existing jailbreak methods by introducing a REINFORCE-based objective that is adaptive, distributional, and semantic. The authors argue convincingly that standard adversarial attacks, which optimize for a fixed affirmative phrase, overestimate LLM robustness by ignoring the full output distribution. The proposed method instead models prompt optimization as a reinforcement learning problem, using policy gradients to maximize the expected reward over harmful completions. Integrated into GCG and PGD attacks, this approach significantly improves ASR, including on models with advanced defenses like Llama 3's circuit-breaker.

The reviewers unanimously appreciate the clarity of writing, methodological soundness, and empirical strength of the work. Notable strengths include the formulation of a semantically grounded objective, robust experiments across models and benchmarks (e.g., HarmBench), and significant performance gains. The authors are responsive in their rebuttals, addressing computational cost concerns, expanding evaluations from 50 to 200 prompts, and justifying decisions regarding baselines, ablations, and the use of LLM-as-a-judge evaluations.

Weaknesses identified include limited discussion on mitigation strategies, no validation of the judge model by human raters, and the computational demands of REINFORCE-based optimization. However, these do not overshadow the paper’s central contribution: demonstrating that non-adaptive attack objectives fundamentally mischaracterize LLM vulnerability and proposing a practically implementable solution with clear empirical benefits.

Overall, this work is timely, methodologically sound, and relevant to both adversarial ML and LLM safety communities. While not a foundational breakthrough, it meaningfully advances the state of adversarial evaluation for LLMs and should be included in the ICML program.